# Cubic and Sphere Magnetic Nanoparticles for Magnetic Hyperthermia Therapy: Computational Results

**DOI:** 10.3390/nano13162383

**Published:** 2023-08-21

**Authors:** Iordana Astefanoaei, Radel Gimaev, Vladimir Zverev, Alexander Tishin, Alexandru Stancu

**Affiliations:** 1Faculty of Physics, Alexandru Ioan Cuza University of Iasi, 700506 Iaşi, Romania; alstancu@uaic.ro; 2Faculty of Physics, M. V. Lomonosov Moscow State University, Moscow 119991, Russiavi.zverev@physics.msu.ru (V.Z.); tishin@amtc.org (A.T.)

**Keywords:** magnetic hyperthermia, cubic/sphere-shaped magnetic nanoparticles (MNPs), hyperthermic temperature values, Comsol Multiphysics, temperature model

## Abstract

Magnetic nanoparticles (MNPs) with various shapes and special (magnetic and thermal) properties are promising for magnetic hyperthermia. The efficiency of this therapy depends mainly on the MNPs’ physical characteristics: types, sizes and shapes. This paper presents the hyperthermic temperature values induced by cubic/sphere-shaped MNPs injected within a concentric tissue configuration (malignant and healthy tissues) when an external time-dependent magnetic field was applied. The space-time distribution of the nanoparticles as a result of their injection within a tumoral (benign/malign) tissue was simulated with the bioheat transport equation (Pennes equation). A complex thermo-fluid model that considers the space-time MNP transport and its heating was developed in Comsol Multiphysics. The cubic-shaped MNPs give a larger spatial distribution of the therapeutic temperature in the tumoral volume compared to the spherical-shaped ones. MNP doses that induce the therapeutic (hyperthermic) values of the temperature (40 ÷ 45 °C) in smaller volumes from the tumoral region were analyzed. The size of these regions (covered by the hyperthermic temperature values) was computed for different magnetite cubic/sphere-shaped MNP doses. Lower doses of the cubic-shaped MNPs give the hyperthermic values of the temperature in a larger volume from the tumoral region compared with the spheric-shaped MNPs. The MNP doses were expressed as a ratio between mass concentration and the maximum clinical accepted doses. This thermo-fluid analysis is an important computational instrument that allows the computations of the MNP doses that give therapeutic temperature values within tissues.

## 1. Introduction

Magnetic hyperthermia is a promising therapy for the treatment of oncological diseases [1,2]. This clinical procedure uses the therapeutic heat generated by magnetic nanoparticles (MNPs) in alternating magnetic fields to destroy abnormal (tumoral/cancerous) tissues in specific conditions [3,4]. The temperature within tissues can be controlled by calculating MNP doses taking into account their thermal and magnetic properties, size and shape [5,6]. The efficiency of this therapeutic method is improved if the MNPs have low concentration and a high value of the specific absorption rate (SAR) [7]. MNPs are intensively investigated to obtain their optimum shape for magnetic hyperthermia applications [8,9,10]. The spatial distribution of MNPs, after the ferrofluid insertion, influences strongly the hyperthermic (therapeutic) values of the temperature within concentric tissues [11,12,13,14,15]. Thermo-fluid analysis that considers the space-time transport and heating of the MNPs within tissues with an intravascular structure is essential for the good design of the magnetic nanoparticle. Recently, many experimental studies have described magnetic nanoparticles (MNPs) with various shapes as potential candidates for magnetic hyperthermia [5,6,16]. These studies should be completed with the analysis of the thermal response of the tissues as a result of their space-time transport and heating in the alternating magnetic fields. 

The goal of this paper is to analyze the capability of cubic/spherical-shaped MNPs to generate and transfer therapeutic heat within tumoral tissues when the alternating magnetic field is applied. The temperature distribution developed by these nanoparticles with various shapes (but the same volume) within a tumor–healthy-tissue configuration was studied. Considering their mass transport as a result of the ferrofluid insertion in the abnormal (tumoral) tissue, the bioheat transfer equation (Pennes equation) was solved using a finite element method (FEM) in the Comsol Multiphysics software 5.6. The MNP doses that determine the hyperthermic values of the temperature were analyzed. The efficiency of these MNPs to destroy the tumoral tissues was discussed. The cubic MNPs have a larger heating rate in a unit time and unit volume than the spherical nanoparticles, ensuring a better heating process and implicitly efficient thermal damage of the abnormal (tumoral) tissues. This comparative study describes the thermal response of the tumoral tissues heated by the cubic and spherical magnetite MNPs.

## 2. Magnetic Hyperthermia with Cubic/Spheric MNPs—Temperature Model

This section describes the temperature model that considers the transport of MNPs and their bioheat transfer within tissues in an applied electromagnetic high-frequency field. In this approach, the intravascular structure of the tumoral (benign/malign) region was considered [17,18]. 

### 2.1. Ferrofluid Injection

Abnormal (tumor) and normal (healthy) tissues are described by concentric tissue configuration with the radius of the tumoral region $R_{1}$ and the radius of the healthy region $R_{2}$. A diluted ferrofluid that contains nanoparticles that have (i) cubic and (ii) spherical shapes with low concentrations (<4% by volumes) was successively injected at the center of the tumoral–healthy geometry (Figure 1). In the intravascular structure of this concentric region, the MNP transport was given by the following equation [3,4,18,19]:
(1)$$\nabla P_{i}=-\frac{\varepsilon_{i}\mu_{i}}{K_{i}}{\vec{v}}_{i}\;\;(i=1,\;2)$$
$\varepsilon_{i}$ and $K_{i}$ are the porosities and permeabilities for abnormal (*i* = 1) and normal (*i* = 2) tissues [20]:$$K_{i}=\frac{{\varepsilon_{i}}^{3}d_{p}^{2}}{180{\left(1-\varepsilon_{i}\right)}^{2},}$$
$P_{i}$ is the interstitial pressure in tissues; $d_{p}=10^{-4}$ m is the value of the mean diameter of the cellular grain in the porous tissues. The MNP velocity ${\vec{v}}_{i}\;$ was computed by solving the following relation [3,20]: (2)$$\nabla \cdot (\varepsilon_{i}{\vec{v}}_{i})=\Phi_{B}^{i}-\Phi_{L}^{i},$$
where the intravascular structure of tissues is described by the expression
(3)$$\Phi_{B}^{i}=\frac{L_{p}S}{V}\left[P_{b}-P_{i}-\sigma_{s}(\pi_{b}-\pi_{i})\right],$$
which is significantly affected by the lymphatic drainage from the normal tissue. The lymphatic term was [3] (Table 1)
(4)$$\Phi_{L}^{i}=\left\{\begin{array}{c} 0,\;\;\;\;\;\;\;\;\;\;\;\;\;\;\;\;\;\;\;\;\;\;\;\;\;\;\;\;\;(i=1)\;\\ \frac{L_{PL}S_{L}}{V}\left[P_{b}-P_{L}\right],\;\;\left(i=2\right). \end{array}\right.$$

MNP velocity as the solution of Equation (2) was solved using the following Dirichlet conditions on the boundaries:

(i) At the top of the needle, MNP velocity depends on the ferrofluid infusion rate, $Q_{v}$ (µL/min):(5)$$U=\frac{Q_{v}}{\pi \;r_{n}^{2}}$$
and the radius of the needle $r_{n}$. (ii) MNP velocity on the boundary of the normal region was zero.

MNP velocity influences strongly its space-time distribution as a result of its convection. The ferrofluid infusion rate, $Q_{v}$ (µL/min), has an important role in MNP transport. When the ferrofluid was inserted within the tumor with a higher value of the infusion rate ($Q_{v})$, MNPs moved at larger distances from the infusion point. 

### 2.2. The Space-Time Distribution of MNPs within Concentric Abnormal (Tumoral)—Normal (Healthy) Tissues 

The space-time distribution of MNPs was computed as the solution of the modified equation, $C_{i}=C_{i}\left(r,t\right)$ for *i* = 1 (abnormal region) and *i* = 2 (normal region) [17,20]:(6)$$\frac{\partial {(\varepsilon }_{i}C_{i})}{\partial t}+\nabla \cdot \left(\vec{v}{\;\varepsilon }_{i\;}C_{i}\right)=\nabla \cdot \left(D_{i}^{*}\nabla C_{i}\right)$$
$$D_{i}^{*}=D_{i,f}^{*}\left[\frac{2\;\varepsilon_{i}}{3-\varepsilon_{i}}\right]$$
where $D_{i,f}^{*}=\frac{\left(1-\lambda^{2}\right)\left(1-2.1044\lambda +2.089\lambda^{3}-0.948\lambda^{5}\right)D_{0}^{i}}{F\tau_{i}}$ are MNP diffusion coefficients in the injected fluid; $\tau_{i}$ are the constant tortuosities: $\frac{1}{\tau_{i}}=1-\frac{2}{3}(1+\varepsilon_{i}){\left(1-\varepsilon_{i}\right)}^{2/3}$; $D_{0}^{i}=\frac{k_{B}T_{i}}{6\pi R\mu }$ are the diffusion coefficients in a bulk liquid medium according to the Stokes–Einstein equation; μ is the viscosity of the fluid containing MNPs; $k_{B}$ is the Boltzmann constant; $T_{i}$ are the temperature values in both (normal and abnormal) regions; and *R* is the radius of MNPs. On the boundaries, the following conditions were imposed: (i) Dirichlet condition ($C_{2}=0$) for $r=R_{2}$; (ii) Neumann condition at the inner normal–abnormal interface:$$C_{1}(r=R_{1})=C_{2}(r=R_{1})\;\mathrm{and}\;D_{1}^{*}{\left.\frac{\partial C_{1}}{\partial r}\right|}_{r=R_{1}}=D_{2}^{*}{\left.\frac{\partial C_{2}}{\partial r}\right|}_{r={\;R}_{1}}$$
(iii) at the center of this concentric geometry (where the fluid was inserted), the mass concentration of the MNPs has a maximum value $C_{1}=C_{max}$. The space-time repartitions (distributions) of MNPs were significantly influenced by (a) the infusion rate described by the parameter *Q_v_*; (b) tissue characteristics such as porosity and permeability; and (c) sizes. In this approach, the volume fractions of MNPs are given by the relation (for *i* = 1 and *i* = 2)
(7)$$\Phi_{i}\left(r\right)=\frac{C_{i}}{\rho_{MNP}}$$
where $\rho_{MNP}$ is MNP mass density. The size of nanoparticles (*R* is the radius for the spherical particles, and *L* is the dimension for the cubic nanoparticles) influences significantly their space-time distribution in tissues. In this approach, cubic/sphere-shaped MNPs were considered to have the same volumes. As a result, the cubic-shaped MNPs with the dimension *L* larger than *R* ($L=R\sqrt[{3}]{\frac{4\pi }{3}}>R$) remain in the vicinity of the ferrofluid insertion point within tissues. The sphere-shaped MNPs have the capability to move at larger distances than the cubic-shaped ones with dimension *L*. This behavior is an important characteristic for this thermo-fluid analysis in magnetic hyperthermia. 

### 2.3. Bioheat Transfer within Tumoral–Healthy Regions

Bioheat transport within the concentric geometry from Figure 1 was studied considering MNPs with the same volumes and various shapes. The space-time temperature distribution in this configuration (abnormal region surrounded by the normal region) was computed by the solutions ${T_{i}=T}_{i}\left(r,t\right),\left(i=1,2\right)$ of the bioheat transport equations (Pennes equations) [17,20]:(8)$$\rho_{i}c_{i}\frac{\partial T_{i}}{\partial t}=\nabla \left[k_{i}\;\nabla T_{i}\right]+\rho_{b}{\;\omega }_{i}^{b}c_{b}\left[T_{art}-T_{i}\right]+Q_{\mathrm{met}}^{i}+Q_{i}$$
where the thermal characteristics for tissues (index *i* = 1, 2) and blood (index “*b*”) are (1) the mass densities (*ρ*_i_) and (*ρ_b_*), (2) specific heat capacities (*c_j_*) and (*c_b_*), (3) thermal conductivities (*k_i_*) and (4) *T_art_* blood temperature and metabolic heat ($Q_{\mathrm{met}}^{i}$) (Table 2). 

The blood perfusion and circulation, thermal conduction and metabolic heat production in living tissues are considered in the above Pennes Equation (8). As a result of the tissue blood perfusion, a cooling effect within tissues appears. This is an important problem to be solved in temperature control in magnetic hyperthermia for cancer therapy. 

(i) On the abnormal–normal region interface, the following boundary conditions were imposed:(9)$$T_{1}\left(r=R_{1}\right)=T_{2}\left(r=R_{1}\right),$$
(10)$$k_{1}{\left.\frac{\partial T_{1}}{\partial r}\right|}_{r=R_{1}}=k_{2}{\left.\frac{\partial T_{2}}{\partial r}\right|}_{r={\;R}_{1}}$$

(ii) On the exterior ($r=R_{2}$) of the normal region (considered sufficiently larger in the computation), the Dirichlet boundary condition was imposed: (11)$$T_{2}\left[r=R_{2}\right]=37\;{}^{\circ}C$$

In Equation (8), $Q_{i}=\Phi_{i}\left(r\right)P\left(\frac{W}{m^{3}}\right)$ is the specific loss power generated by the volume fraction $\Phi_{i}\left(r\right)$ of MNPs, within the volume of the concentric (abnormal–normal) regions in the presence of the applied external field. 

It can be observed that the spatial distribution of the MNPs $\Phi_{i}\left(r\right)$ given by the relations (7) influences significantly the temperature values within normal–abnormal (tumoral/healthy) tissues. The specific loss power density developed by a nanoparticle in the external magnetic field, $P_{\;}$, is given by the following relation:(12)$$P_{\;\;}=\mu_{0}f\pi \chi ‘‘H^{2}\;\;\mathrm{and}\;\;\chi ‘‘=\frac{\mu_{0}M_{s}^{2}V}{{3k}_{B}T}\frac{2\pi f\tau }{1+{\left(2\pi f\tau \right)}^{2}}$$

*H*(kA/m) and *f*(kHz) define the amplitude and frequency of the magnetic field. The effective relaxation time, $(\tau =\frac{\tau_{N\;}\tau_{B\;}}{\tau_{N}+\tau_{B\;}}$), depends on the Brown relaxation time $\left(\tau_{B}=\frac{{3\;\eta \;V}_{H}\;}{k_{B}\;T}\right)$ and Néel relaxation time $\tau_{N}=\tau_{0}\frac{\sqrt{\pi }}{2}\frac{exp\left[\frac{K\;V}{k_{B}T}\right]}{\sqrt{\frac{K\;V}{k_{B}T}}}$. η is the liquid viscosity, $V_{H}=V{\left(1+\frac{\delta }{R}\right)}^{3}$ is MNP hydrodynamic volume, and δ is the surfactant layer thickness. The specific loss power density of the MNPs depends on (i) the magnetic properties: saturation magnetization *M_s_* and anisotropy constant *K*, (ii) amplitude and the frequency of the magnetic field and (iii) MNP characteristics: size and the volume fraction of the MNPs. A high value of saturation magnetization involves a high value of magnetic susceptibility and implicitly a better response of the nanoparticle in the external field applied. 

Irreversible transformations appear within abnormal (benign/malign) tissues when the temperature increases to the hyperthermic level of 40 ÷ 45 °C as a result of the therapeutic heat delivered by the MNPs when the magnetic fields are applied. The heating processes in this temperature range determine the thermal damage of the abnormal tissues due to their low thermal resistance. These thermal damage processes start with the denaturation of the proteins. The undamaged fraction of abnormal tissues depending on the temperature was defined according to the following relations [20,21,22]:(13)$$\theta_{i}(r,t)=\exp\left[-\Omega_{i}(r,t)\right]\left(i=1,2\right)$$
where the adimensional index $\Omega_{i}$ (which describes the thermal damage of the tissues) is defined by the Arrhenius law [15]:$$\Omega_{i}(r,t)=\int_{0}^{t}A_{i}exp\left(-\frac{E_{a}^{i}}{RT_{i}}\right)d\tau$$

$E_{a}^{i}\left[J\cdot {mol}^{-1}\right]$ is the activation energy, $A_{i}$ [s^−1^] is the frequency factor, and ${R[Jmol}^{-1}K^{-1}$] is the universal gas constant. The undamaged fraction of tissues (13) $\theta_{i}\left(r,t\right)$ decreases from the constant value $\theta_{i}=1$ (at the beginning of the heating process when the abnormal tissue is completely undamaged) to the constant value $\theta_{i}=0$ (at the end of the heating process), when the abnormal tissue is considered completely damaged. In the bioheat transport Equation (8), the blood perfusion rate is considered strongly dependent on the thermal damage index $\Omega_{i}$ according to the relation $\omega_{i}^{b}=\omega_{i}^{0}\;exp(-\Omega_{i}\left(r,t\right))$ [11]. 

## 3. Results and Discussion

The temperature values within a configuration of the liver tissues were determined in the following range for frequency f: 100–650 kHz and amplitude H: 1–7 kA/m of the magnetic field. The discretization of the concentric tumoral–healthy tissue geometries was performed using the finite element method (FEM) in the Comsol Multiphysics software. The solutions of Equations (1), (6) and (8) were obtained using extremely fine mesh which contains 50,000 domain elements. The time-dependent solver with the implicit backward differentiation (BDF) method was used. This is an implicit solver that uses backward differentiation formulas with a variable discretization order and automatic step-size selection. For the quality of the solutions, higher-order schemes are used. The variation of the time step in the course of the simulation can be followed in the Convergence Plot:

1. First, the temperature generated by a magnetite nanoparticle with a cubic shape was compared with the one with a spherical shape when a magnetic field was applied. In this idealized model, both magnetite nanoparticles with the same volumes were inserted within a concentric abnormal–normal cell configuration (Figure 2a,b). In the same magnetic field conditions, the cubic nanoparticle induces a higher temperature rise than the spherical one.

In this analysis, the spherical magnetic nanoparticles have the radius R = 8 ÷ 10 nm, and the length of the cubic magnetic nanoparticles (with the same volumes) is $L=R\;\sqrt[{3}]{\frac{4\pi }{3}}\approx 13÷16\;nm.$ The size of the sphere nanoparticles can influence their magnetic anisotropy. This is a fundamental magnetic property, which refers to the preferential alignment of the magnetic moments in a particular direction. In the spherical nanoparticles, the anisotropy arises from the shape anisotropy, which results from the difference in energy associated with aligning the magnetic moment parallel or perpendicular to the surface of the nanoparticle.

As the size of the spherical nanoparticle decreases, the anisotropy energy associated with aligning the magnetic moment perpendicular to the surface of the nanoparticle increases. This can lead to a higher magnetic anisotropy in smaller nanoparticles. Furthermore, smaller nanoparticles can exhibit single-domain magnetic behavior, which means that all of the magnetic moments in the nanoparticle are aligned in the same direction. This can result in a higher magnetic anisotropy due to the absence of domain walls. In cubic nanoparticles, the anisotropy arises from the shape anisotropy and the crystallographic anisotropy, which results from the difference in energy associated with aligning the magnetic moment along different crystallographic directions. As the size of the cubic nanoparticle decreases, the shape anisotropy energy associated with aligning the magnetic moment perpendicular to the surface of the nanoparticle increases, similar to what happens in spherical nanoparticles. In addition, the magnetocrystalline anisotropy energy can also change with the nanoparticle size due to the quantization of electronic energy levels, which can result in different electronic structures for different sizes. Overall, the combination of shape and magnetocrystalline anisotropies can lead to a complex dependence of magnetic anisotropy on nanoparticle size, and the optimal size for magnetic hyperthermia may depend on the specific material and crystalline structure. In general, smaller cubic nanoparticles can exhibit higher anisotropy, which can make them more efficient in magnetic hyperthermia.

2. Second, the spatial distribution of the temperature generated by the (spherical and cubic) nanoparticles with the same values of the specific loss power P in a concentric abnormal–normal (tumor–healthy) cell was analyzed. Figure 3 shows the spatial temperature values developed by these two nanoparticles in a radial direction within a malignant cell. In this case, a system of coordinates Oxyz was considered at the center of the cell with a 100 nm diameter. This simple temperature analysis shows that the cubic symmetry gives a larger spatial distribution of the temperature compared with the spherical one. 

In magnetic hyperthermia, the surface area and the gap between the magnetic nanoparticles (MNPs) and the surrounding tissue have a significant impact on the heating efficiency and effectiveness of the treatment. The surface area is an important factor, as it determines the amount of energy that can be absorbed by the MNPs when subjected to an external alternating magnetic field. MNPs with a larger surface area have a greater number of magnetic moments exposed to the magnetic field, which leads to greater heat generation. Thus, MNPs with a larger surface area, like cubic magnetic nanoparticles, are generally more effective for hyperthermia treatment. Also, the gap between the MNPs and the tissue is also an important factor to consider in magnetic hyperthermia. The distance between the MNPs and the target tissue can affect the heating efficiency, as it determines the amount of heat that can be transferred from the MNPs to the surrounding tissue. A smaller gap between the MNPs and the tissue can result in more efficient heating, as the heat generated by the MNPs is more readily transferred to the tissue. 3. The values of the specific loss power density developed by the cubic- and sphere-shaped nanoparticle with the frequency and amplitude of the magnetic field were studied. The values of saturation magnetization M_s_ and anisotropy constant K were considered from published experimental data [9,10]. The cubic shape of the magnetic particle releases a larger energy per unit time in the volume unit when the frequency and amplitude of the magnetic field are changed (Figure 4a,b). An important difference between the values of the specific loss power density generated by the spherical and cubic shapes is presented in Figure 4c. The temperature values were strongly influenced by the field dependence (Figure 4d). The temperature field generated by the cubic-shaped MNPs reaches the therapeutic range for a lower value of the amplitude of the magnetic field (H = 3 ÷ 5 kA/m). The shapes of the MNPs influence strongly the temperature values within the abnormal region. This important thermal behavior appears as a result of both magnetic nanoparticle characteristics: (i) magnetic properties: the saturation magnetization M_s_ and anisotropy constant K; and (ii) the different sizes of the nanoparticles with the same volumes used in these simulations.

Cubic-shaped magnetic nanoparticles release more energy per unit volume compared to spherical ones. From the magnetic point of view, a simple explanation of this observation is given next. The release of energy in magnetic nanoparticles depends on their magnetic properties, specifically the process of magnetization and demagnetization. The energy released during this process is known as hysteresis loss or magnetic energy loss. Considering the difference in shape between cubic and spherical magnetic nanoparticles, a cubic shape has more edges and corners compared to a sphere. These edges and corners introduce discontinuities in the crystal lattice structure of the nanoparticle. Such irregularities can enhance the energy loss during magnetization and demagnetization processes. During magnetization, when an external magnetic field is applied, the magnetic moments of the nanoparticles align with the field. This alignment requires energy. The energy required to align the magnetic moments is stored in the form of magnetic potential energy within the nanoparticles. However, when the external magnetic field is removed, the nanoparticles return to their original state, and this demagnetization process also releases energy. The energy release is due to the relaxation of the magnetic moments and the conversion of magnetic potential energy back into other forms (such as thermal energy). Regarding the differences between cubic and spherical nanoparticles, the edges and corners in cubic nanoparticles act as regions with higher magnetic field gradients compared to the smooth surface of spherical nanoparticles. These gradient variations create additional strain and increase the energy required for magnetization. During the demagnetization process, the irregularities in the cubic nanoparticles lead to a higher probability of domain wall motion and nucleation, resulting in more energy release compared to spherical nanoparticles. The shape of cubic nanoparticles provides more favorable sites for domain wall movement and facilitates the relaxation of magnetic moments, thereby increasing the energy release. In summary, the cubic shape of magnetic nanoparticles, with its edges and corners, introduces crystal lattice irregularities that enhance the energy loss during magnetization and demagnetization processes. This results in a higher energy release per unit volume compared to spherical nanoparticles. This explanation provides a simplified understanding of the phenomenon, and the actual behavior of magnetic nanoparticles is more complex, involving factors such as material composition, particle size and surface effects. 

Particle size influences the effect of energy release in magnetic nanoparticles. The size of the nanoparticles can significantly influence their magnetic properties, including the energy release during magnetization and demagnetization processes. As the size of the nanoparticles decreases, several size-dependent effects become active. One important effect is the increase in the surface-to-volume ratio. When nanoparticles are smaller, a larger proportion of their atoms or ions are located at the surface compared to the bulk. The surface atoms have different characteristics and can exhibit enhanced reactivity and altered magnetic behavior compared to the interior atoms. In the case of energy release, the surface effects can affect the magnetization and demagnetization processes. Smaller nanoparticles have a higher proportion of atoms located at the surface, which can result in higher surface energy and increased magnetic anisotropy. Magnetic anisotropy refers to the preferred orientation of the magnetic moments within the nanoparticles. The increased magnetic anisotropy in smaller nanoparticles can lead to more stable magnetic configurations and a higher energy barrier for magnetization. As a result, smaller nanoparticles may require more energy to achieve magnetization, and consequently, they can release more energy during the demagnetization process. Furthermore, the size-dependent effects also influence the dynamics of domain wall motion and nucleation. In smaller nanoparticles, the domain walls have to traverse a shorter distance, making their movement easier and more energetically favorable. This can lead to a higher probability of domain wall motion and nucleation, enhancing the energy release. It is worth noting that while smaller nanoparticles tend to exhibit enhanced energy release, there is a limit to this effect. At extremely small sizes, quantum confinement effects become dominant, and the behavior of the nanoparticles deviates from the classical understanding. In such cases, additional factors such as quantum tunneling and spin-crossover phenomena come into play, leading to more complex magnetic behavior. In summary, particle size plays a crucial role in the energy release of magnetic nanoparticles. Smaller nanoparticles with an increased surface-to-volume ratio can exhibit enhanced magnetic anisotropy, more favorable domain wall dynamics and higher energy release during magnetization and demagnetization processes. However, it is important to consider the interplay of various factors, including shape, composition and surface effects, to fully understand the behavior of magnetic nanoparticles.

In the following sections, one shows the temperature analysis for two concentric domains (tissues) (the abnormal and normal regions) (Figure 1).

3. In the following simulations, the space-time temperature values generated by the same magnetite MNP doses that had the cubic and spherical shapes in a magnetic field (f = 200 kHz, H = 5 kA/m) were computed from Equations (1), (6) and (8) for a malignant tissue that had a 40 mm diameter. At the center of the tumor–healthy-tissue configuration, the maximum clinical accepted concentration was C_max_ = 10 mg/cm^3^. 

The thermal damage of the tumoral tissue was studied in the same magnetic field conditions for both types of MNP shapes. Cubic-shaped MNPs have a larger spatial distribution which determines a larger volume covered by the hyperthermic temperature values compared to the spherical ones (Figure 5a). In this case, a higher percentage of the tumoral tissue was destroyed, as shown in Figure 5b. Both Figure 5a,b describe the temperature and thermal damage induced by the MNPs with cubic and sphere shapes. The cubic symmetry gives a better spatial distribution of the MNPs and implicitly a better space-time temperature distribution (Figure 5a). In this case, the malignant tissue is thermally damaged more efficiently (Figure 5b). Cubic MNPs are more efficient in the bioheat transport within normal–abnormal tissues. In the following, in the volume of the tumoral tissue, a sphere (“therapeutic sphere”) that contains at every point the therapeutic (hyperthermic) temperature values of 40 ÷ 45 °C was delimited. The size (the radius $r_{0}$) of this therapeutic region (sphere) depends strongly on the MNP doses. Figure 6 shows the evolution of the “therapeutic sphere” size (the parameter $r_{0}\;$) (covered by the therapeutic temperature values) with MNP doses (expressed as a ratio between particle concentrations and maximum clinical accepted concentration). The values of the parameter $r_{0}$ increase with the MNP doses for every symmetry. Lower cubic-shaped MNP doses determine a larger “therapeutic sphere” from the tumoral volume compared with the spherical ones (Figure 6). 

## 4. Conclusions

The shape of MNPs is very important for the efficiency of the magnetic hyperthermia method. Cubic-shaped magnetic nanoparticles release larger energy in unit volume per unit time compared with the spherical ones. A similar trend of heating efficiency in cubic magnetic nanoparticles was observed in the experimental work published [10]. Low concentrations of the cubic-shaped MNPs burn a larger volume from the tumoral tissue. In this case, the hyperthermic temperature values in the abnormal region were reached at lower values for the frequency and amplitude of the magnetic field. These magnetic nanoparticles can be used in magnetic hyperthermia more efficiently. Higher MNP doses with cubic and spherical shapes determine an increase in the region from the tumoral volume which contains the therapeutic temperature values.

## Figures and Tables

**Figure 1 nanomaterials-13-02383-f001:**
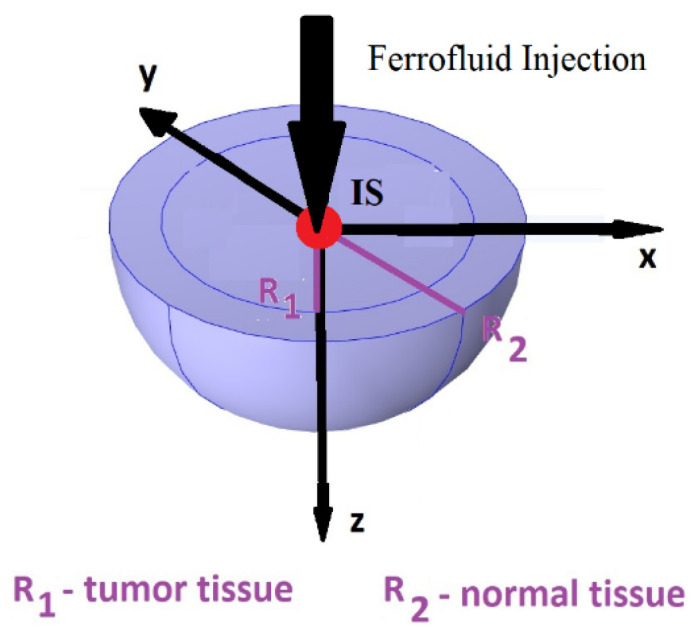
Concentric abnormal (tumoral)—normal (healthy) tissue configuration.

**Figure 2 nanomaterials-13-02383-f002:**
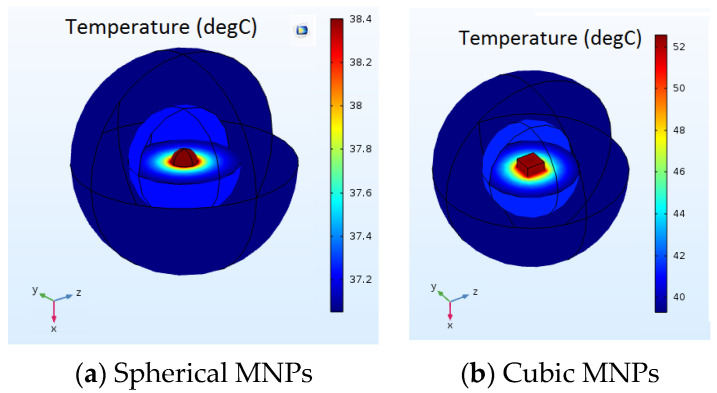
Temperature generated by the MNPs with (**a**) spherical and (**b**) cubic shape in the applied magnetic field.

**Figure 3 nanomaterials-13-02383-f003:**
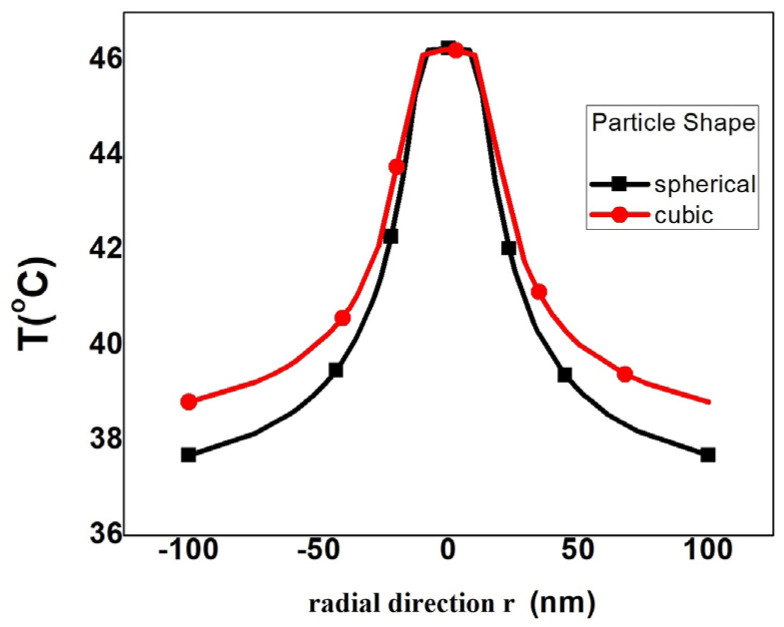
The radial temperature for nanoparticles with spheric and cubic shape.

**Figure 4 nanomaterials-13-02383-f004:**
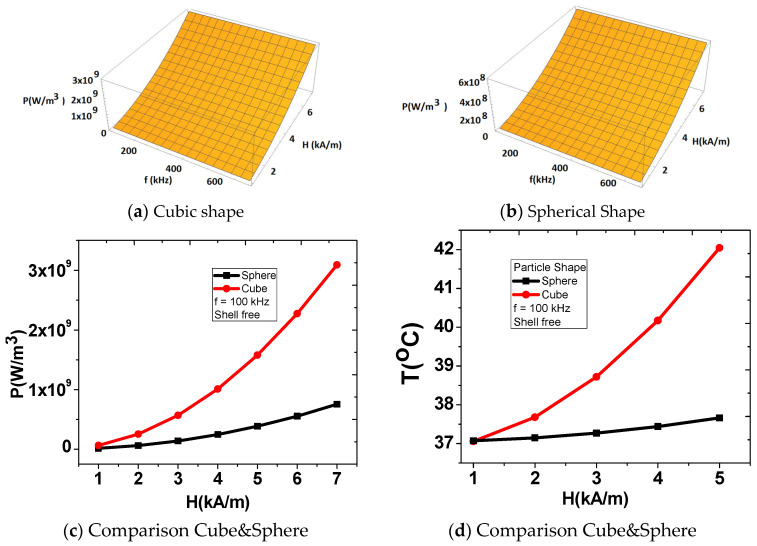
The dependence of the specific loss power density P on frequency and amplitude of the magnetic field for a cubic- and sphere-shaped nanoparticle.

**Figure 5 nanomaterials-13-02383-f005:**
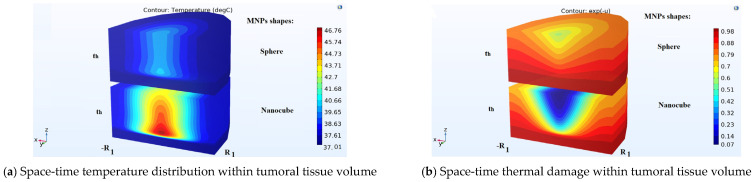
Temperature (**a**) and thermal damage (**b**) given by the spherical and cubic-shaped magnetic nanoparticles.

**Figure 6 nanomaterials-13-02383-f006:**
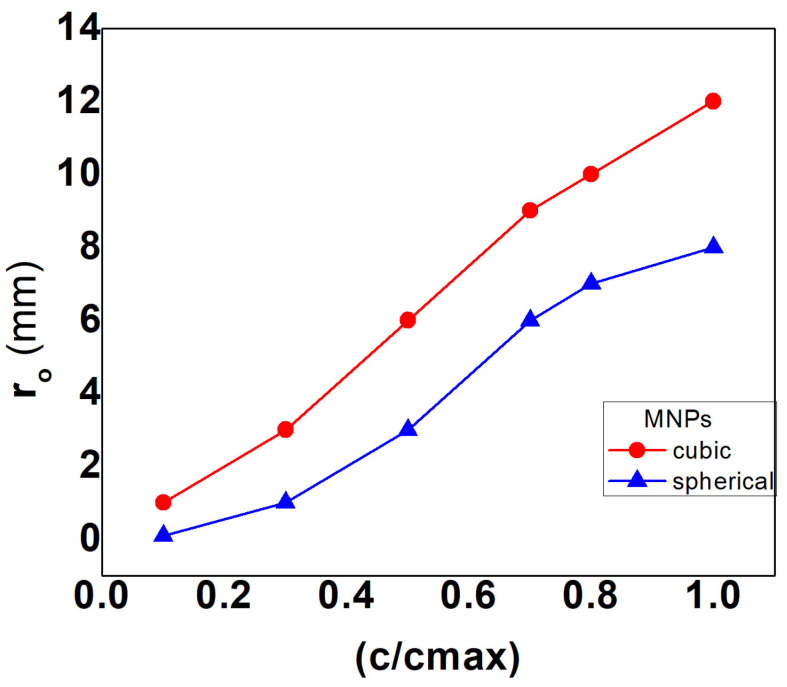
The radius of “the therapeutic sphere” and cubic/spheric-shaped MNP doses. “The therapeutic sphere” is covered by the therapeutic (hyperthermic) temperature values given by the various MNP doses in applied magnetic fields.

**Table 1 nanomaterials-13-02383-t001:** Numeric values of the parameters [12,14,15].

Vascular Characteristics	Abnormal Region (*i* = 1)	Normal Region (*i* = 2)
L_P_ (cm $mm{Hg}^{-1}s^{-1}$ ) hidraulic conductivity	2.8 × 10^−7^	0.36 × 10^−7^
L_PV_ S_L_V^−1^ ($mm{Hg}^{-1}s^{-1}$) lymphatic coefficient	-	5 × 10^−5^
P_b_ (mmHg) static blood pressure	15.6	15.6
π_b_ (mmHg) plasma protein pressure	-	-
π_i_ (mmHg) the interstitial pressure	15	10
σ	0.82	0.91
S_V_V^−1^ (${\mathrm{cm}}^{-1}$)		
P_L_	-	0

**Table 2 nanomaterials-13-02383-t002:** The values of the constants [20].

Characteristics	Magnetite	Abnormal Region	Normal Region	Blood
Mass densities $\left(kg/m^{3}\right)$	5180	1160	1060	1000
Specific heat capacities (J/Kg K)	670	3600	3600	4180
Thermal conductivities (W/mK)	40	0.4692	0.512	-
Metabolic heat $Q_{\mathrm{met}}^{i}$ (W/m^3^)	-	5790	700	-
Blood perfusion rate (1/s)	-	0.0064	0.0064	-
Frequency range f(kHz)	100–650	-	-	-
Magnetic field amplitude H(kA/m)	0–15	-	-	-

## Data Availability

The authors declare that all data supporting the findings of this study are available within the article.

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
