# Peer review of "Cubic and Sphere Magnetic Nanoparticles for Magnetic Hyperthermia Therapy: Computational Results"

_nanomaterials, 2023, doi:10.3390/nano13162383_

Round 1
Reviewer 1 Report
The manuscript by Alexandru Stancu et al. explored the cubic and sphere magnetic nanoparticles. Particularly, they use thermo-fluid analysis to study the system. Overall, this work is very interesting and could attract attention from researchers in the field. I'd like to recommend the acceptance of this work after minor revision.
1. Space-Time thermal Damage was shown in Figure 5 for cubic and sphere magnetic nanoparticles. When the authors perform the analysis, what is the size of cubic and sphere nanoparticles.
2. Besides shape difference, how would the size of sphere nanoparticles influence the magnetic properties?
Author Response
Answers to the reviewer 1
Manuscript Title: " Cubic and Sphere shaped Magnetic Nanoparticles for Magnetic Hyperthermia Therapy: Iordana Astefanoaei, Radel Gimaev, Vladimir Zverev, Alexander Tishin , Alexandru Stancu.
Thank you so much for your kind recommendation concerning revising of this paper, for the suggestions and recommendations. In order to improve our manuscript, we revised the paper according with your advices. The all supplementary phrases added are presented inside the paper.
Question 1.
- Space-Time thermal Damage was shown in Figure 5 for cubic and sphere magnetic nanoparticles. When the authors perform the analysis, what is the size of cubic and sphere nanoparticles.
Answer 1. The following comment was inserted within paper (page 6): `In this analysis, the spherical magnetic nanoparticles have the radius R = 8 ÷ 10 nm and the length of the cubic magnetic nanoparticles (with the same volumes) is .`
Question 2.
- Besides shape difference, how would the size of sphere nanoparticles influence the magnetic properties?
Answer 2. The following comment was inserted in the paper on the page 6:
The size of the sphere nanoparticles can influence their magnetic anisotropy. This is a fundamental magnetic property – which refers to the preferential alignment of the magnetic moments in a particular direction. In the spherical nanoparticles, the anisotropy arises from the shape anisotropy, which results from the difference in energy associated with aligning the magnetic moment parallel or perpendicular to the surface of the nanoparticle. As the size of the spherical nanoparticle decreases, the anisotropy energy associated with aligning the magnetic moment perpendicular to the surface of the nanoparticle increases. This can lead to a higher magnetic anisotropy in smaller nanoparticles. Furthermore, smaller nanoparticles can exhibit single-domain magnetic behavior, which means that all of the magnetic moments in the nanoparticle are aligned in the same direction. This can result in a higher magnetic anisotropy due to the absence of domain walls. In cubic nanoparticles, the anisotropy arises from the shape anisotropy and the magnetocrystalline anisotropy, which results from the difference in energy associated with aligning the magnetic moment along different crystallographic directions. As the size of the cubic nanoparticle decreases, the shape anisotropy energy associated with aligning the magnetic moment perpendicular to the surface of the nanoparticle increases, similar to what happens in spherical nanoparticles. In addition, the magnetocrystalline anisotropy energy can also change with the nanoparticle size due to the quantization of electronic energy levels, which can result in different electronic structures for different sizes. Overall, the combination of shape and magnetocrystalline anisotropies can lead to a complex dependence of magnetic anisotropy on nanoparticle size, and the optimal size for magnetic hyperthermia may depend on the specific material and crystalline structure. In general, smaller cubic nanoparticles can exhibit higher anisotropy, which can make them more efficient in magnetic hyperthermia.
Thank you for your consistent recommendations and re-consideration.

Reviewer 2 Report
This is a simulation/calculation work, which discussed the impact of sizes and shapes of magnetic nanoparticles (MNPs) on their magnetic and thermal properties. The cubic shaped MNPs are better distributors of therapeutic temperature compared with spherical ones. Generally, it was not clear what MNPs were used in these simulations and there was no perspective comparison with experimental results.
1. the fact that this is a simulation study of an idealized model should be mentioned in the title.
2. The kind of MNPs was not clear, is the simulation based on maghemite or magnetite? There was no mention of the sizes of these MNPs when compared with experimental ones (a table should be given with data comparison), the factors like surface coating and aggregation should take into consideration.
3. Cubic and sphere MNPs with the same volumes were used in the simulation, and the impact of surface area, and the gap between MNPs and tissue should not be ignored.
4. Typos such as ‘terapeutic’, ’40 ÷ 45’, ‘ferofluid’ and others
typos need to be corrected
Author Response
Answers to the reviewer 2
Manuscript Title: "Cubic and Sphere shaped Magnetic Nanoparticles for Magnetic Hyperthermia Therapy: Iordana Astefanoaei, Radel Gimaev, Vladimir Zverev, Alexander Tishin , Alexandru Stancu.
Thank you so much for your kind recommendation concerning revising of this paper, for the suggestions and recommendations. In order to improve our manuscript, we revised the paper according with your advices. The all supplementary phrases added are presented inside the paper.
This is a simulation/calculation work, which discussed the impact of sizes and shapes of magnetic nanoparticles (MNPs) on their magnetic and thermal properties. The cubic shaped MNPs are better distributors of therapeutic temperature compared with spherical ones. Generally, it was not clear what MNPs were used in these simulations and there was no perspective comparison with experimental results.
Question 1.
- the fact that this is a simulation study of an idealized model should be mentioned in the title.
Answer 1. We improved this point. Now, the title of this paper becomes: "Cubic and Sphere shaped Magnetic Nanoparticles for Magnetic Hyperthermia Therapy. Computational Results.
Question 2. The kind of MNPs was not clear, is the simulation based on maghemite or magnetite? There was no mention of the sizes of these MNPs when compared with experimental ones (a table should be given with data comparison), the factors like surface coating and aggregation should take into consideration.
Answer 2. In this paper, the magnetite nanoparticles were used in simulations. We added in the paper on page 6, this point. The following comment was inserted within paper (page 6): `In this analysis, the spherical magnetic nanoparticles have the radius R = 8 ÷ 10 nm and the length of the cubic magnetic nanoparticles (with the same volumes) is .` This paper considers the magnetic nanoparticles with small sizes. In this approach, their aggregation was not studied. The size of the magnetic nanoparticles plays a significant role in their aggregation behavior. Smaller nanoparticles have higher Brownian motion due to their lower mass which makes them to be more dispersed. Also, by steric stabilization – a polymer surfactants can be attached to the nanoparticle surface, creating a repulsive barrier – preventing close contact between particles. The aggregation can be controlled by the PH and ionic strength of the ferrofluid and nanoparticles concentrations. Larger nanoparticles tend to aggregate more easily. Due to this reason, in this approach of smaller nanoparticle sizes, the aggregation was not considered. But, this case of larger sizes of the nanoparticles with aggregation involve more theoretical and computational attention. This case will be described in another work.
Question 3. Cubic and sphere MNPs with the same volumes were used in the simulation, and the impact of surface area, and the gap between MNPs and tissue should not be ignored.
Answer 3. We added an explanation on this point in the paper:
In magnetic hyperthermia, the surface area and the gap between the magnetic nanoparticles (MNPs) and the surrounding tissue have a significant impact on the heating efficiency and effectiveness of the treatment. The surface area is an important factor – as it determines the amount of energy that can be absorbed by the MNPs when subjected to an external alternating magnetic field. MNPs with a larger surface area, have a greater number of magnetic moments exposed to the magnetic field, which leads to greater heat generation. Thus, MNPs with a larger surface area, like cubic magnetic nanoparticles are generally more effective for hyperthermia treatment.
Also the following comment was added in this paper on page 8:
Also, the gap between the MNPs and tissue is also an important factor to consider in magnetic hyperthermia. The distance between the MNPs and the target tissue can affect the heating efficiency, as it determines the amount of heat that can be transferred from the MNPs to the surrounding tissue. A smaller gap between the MNPs and the tissue can result in more efficient heating, as the heat generated by the MNPs is more readily transferred to the tissue.
Question 4.
- Typos such as ‘terapeutic’, ’40 ÷ 45’, ‘ferofluid’ and others
Answer 4. We solved this point.
Thank you for your consistent recommendations and re-consideration.

Reviewer 3 Report
Recent interest to magnetic nanoparticles has increased due to their numerous applications in diagnostics and therapy. A promising method of therapy - magnetic hyperthermia in practice encounters many problems. For example, temperature inhomogeneity or taking into account the shape of particles leads to difficulties in designing the expected thermal effect. One of the ways to overcome these difficulties is the numerical simulation. The authors performed such simulations and found that cubic magnetic nanoparticles release more energy per unit volume during certain time compared to spherical ones. If this numerical find is reliable, this is an interesting progress and the work is worthy of publication. However, before final acceptance, the authors should take into account a number of comments and answer a few questions.
1. I would be desirable to see a simple physical explanation of the main numerical observation: “cubic-shaped magnetic nanoparticle release more energy per unit volume compared to spherical ones”. Does particle size matter in this effect?
2. I did not find information in the text about the size of the particle used in thermophysical calculations. This needs to be added even if the answer to the last part of question 1 is no.
3. The abstract states " This important result was in good agreement with the experimental ones recently published in literature". Unfortunately, I did not find in the text a reference to a specific experimental work in close connection with this statement. Add it. Probably, the authors meant Ref 10. However, I have doubts about the comparison, because the calculation was done on a single particle, and the experiment was on an ensemble of particles in a non-zero field. The thermophysical behavior of the ensemble has a lot of features. Add discussion to the text about this.
4. Lines 195 for temperature analysis mention "concentric domains with diameter of 40 mm and 100 mm, respectively" followed by "cell having 100nm diameter". Check it.
5. With such an abundance of equations in the text, they should be drawn up more accurately. In particular, try to stick to a unified style. For example, in line 129 the same values are typed in italics and without it. In line 166 Ms must be subscripted. Correct it throughout the text.
6. Unify the character used to denote a range. Now in the text and tables there is a mixture of characters ÷ (in my opinion correct), - and —.
7. Figures 3 and 6 need to be improved and clarified.
7.1 There are no axes ticks, this makes the presentation of numerical information careless.
7.2 What is the meaning of the symbols in Figure 3? If these are the values where data was taken, why are they connected by a smooth line with a break feature around 10 nm where no data was taken?
7.3 For a cubic particle, the thermal front will not be completely spherical. What then is the meaning of r in Fig. 3 and r0 in Fig. 6? Is it correct to call this value the radius, because for a non-spherical front it will be different in different directions?
8. Results obtained using a finite element method (FEM) in Comsol Multiphysics software. To assess the reliability of the obtained data, information about the finite element mesh is critical. This information needs to be added.
Author Response
Answers to the reviewer 3
Manuscript Title: " Cubic and Sphere shaped Magnetic Nanoparticles for Magnetic Hyperthermia Therapy: Iordana Astefanoaei, Radel Gimaev, Vladimir Zverev, Alexander Tishin , Alexandru Stancu.
Thank you so much for your kind recommendation concerning revising of this paper, for the suggestions and recommendations. In order to improve our manuscript, we revised the paper according with your advices. The all supplementary phrases added are presented inside the paper.
Recent interest to magnetic nanoparticles has increased due to their numerous applications in diagnostics and therapy. A promising method of therapy - magnetic hyperthermia in practice encounters many problems. For example, temperature inhomogeneity or taking into account the shape of particles leads to difficulties in designing the expected thermal effect. One of the ways to overcome these difficulties is the numerical simulation. The authors performed such simulations and found that cubic magnetic nanoparticles release more energy per unit volume during certain time compared to spherical ones. If this numerical find is reliable, this is an interesting progress and the work is worthy of publication. However, before final acceptance, the authors should take into account a number of comments and answer a few questions.
Question 1. I would be desirable to see a simple physical explanation of the main numerical observation: “cubic-shaped magnetic nanoparticle release more energy per unit volume compared to spherical ones”. Does particle size matter in this effect?
Answer 1 We added this explanation in paper:
Cubic-shaped magnetic nanoparticles release more energy per unit volume compared to spherical ones. This release of energy in magnetic nanoparticles, depends on their magnetic properties, specifically the process of magnetization and demagnetization. The energy released during this process is known as hysteresis loss or magnetic energy loss. If it is considered the difference in shape between cubic and spherical magnetic nanoparticles, a cubic shape has more edges and corners compared to a sphere. These edges and corners introduce discontinuities in the crystal lattice structure of the nanoparticle. Such irregularities can enhance the energy loss during magnetization and demagnetization processes. During magnetization, when an external magnetic field is applied, the magnetic moments of
the nanoparticles align with the field. This alignment requires energy. The energy required to align the magnetic moments is stored in the form of magnetic potential energy within the nanoparticles. However, when the external magnetic field is removed, the nanoparticles return to their original state, and this demagnetization process also releases energy. The energy release is due to the relaxation of the magnetic moments and the conversion of magnetic potential energy back into other forms (such as thermal energy). Now, let's consider the differences between cubic and spherical nanoparticles. The edges and corners in cubic nanoparticles act as regions with higher magnetic field gradients compared to the smooth surface of spherical nanoparticles. These gradient variations create additional strain and increase the energy required for magnetization. During the demagnetization process, the irregularities in the cubic nanoparticles lead to a higher probability of domain wall motion and nucleation, resulting in more energy release compared to spherical nanoparticles. The irregular shape of cubic nanoparticles provides more favorable sites for domain wall movement and facilitates the relaxation of magnetic moments,
thereby increasing the energy release. In summary, the cubic shape of magnetic nanoparticles, with its edges and corners, introduces crystal lattice irregularities that enhance the energy loss during magnetization and demagnetization processes. This results in a higher energy release per unit volume compared to spherical nanoparticles. This explanation provides a simplified understanding of the phenomenon, and the actual behavior of magnetic nanoparticles is more complex, involving factors such as material composition, particle size, and surface effects.
Regarding the size effect, we added in the paper, the following comment:
Particle size influences the effect of energy release in magnetic nanoparticles. The size of the nanoparticles can significantly influence their magnetic properties, including the energy release during magnetization and demagnetization processes.
As the size of the nanoparticles decreases, several size-dependent effects becomes active. One important effect is the increase in surface-to-volume ratio. When nanoparticles are smaller, a larger proportion of their atoms or ions are located at the surface compared to the bulk. The surface atoms have different characteristics and can exhibit enhanced reactivity and altered magnetic behavior compared to the
interior atoms. In the case of energy release, the surface effects can affect the magnetization and demagnetization processes. Smaller nanoparticles have a higher proportion of atoms located at the surface, which can result in higher surface energy and increased magnetic anisotropy. Magnetic anisotropy refers to the preferred orientation of the magnetic moments within the nanoparticles. The increased magnetic anisotropy in smaller nanoparticles can lead to more stable
magnetic configurations and a higher energy barrier for magnetization. As a result, smaller nanoparticles may require more energy to achieve magnetization, and consequently, they can release more energy during the demagnetization process. Furthermore, the size-dependent effects also influence the dynamics of domain wall
motion and nucleation. In smaller nanoparticles, the domain walls have to traverse a shorter distance, making their movement easier and more energetically favorable.
This can lead to a higher probability of domain wall motion and nucleation,
enhancing the energy release. It's worth noting that while smaller nanoparticles tend to exhibit enhanced energy release, there is a limit to this effect. At extremely small sizes, quantum confinement effects become dominant, and the behavior of the nanoparticles deviates from the classical understanding. In such cases, additional factors such as quantum tunneling and spin-crossover phenomena come into play, leading to more complex magnetic behavior. In summary, particle size plays a crucial role in the energy release of magnetic nanoparticles. Smaller nanoparticles with increased surface-to-volume ratio can exhibit enhanced magnetic anisotropy, more favorable domain wall dynamics, and higher energy release during magnetization and demagnetization processes. However, it's important to consider the interplay of various factors, including shape, composition, and surface effects, to fully understand the behavior of magnetic nanoparticles.
Question 2. I did not find information in the text about the size of the particle used in thermophysical calculations. This needs to be added even if the answer to the last part of question 1 is no.
Answer 2. The following comment was inserted within paper (page 6): `In this analysis, the spherical magnetic nanoparticles have the radius R = 8 ÷ 10 nm and the length of the cubic magnetic nanoparticles (with the same volumes) is .`
Question 3. The abstract states "This important result was in good agreement with the experimental ones recently published in literature". Unfortunately, I did not find in the text a reference to a specific experimental work in close connection with this statement. Add it. Probably, the authors meant Ref 10. However, I have doubts about the comparison, because the calculation was done on a single particle, and the experiment was on an ensemble of particles in a non-zero field. The thermophysical behavior of the ensemble has a lot of features. Add discussion to the text about this.
Answer 3. The statement "This important result was in good agreement with the experimental ones recently published in literature", was removed from abstract.
Indeed, a similar trend of heating efficiency (like in ref. 10) in cubic magnetic nanoparticles was observed. This aspect was improved and mentioned in the paper.
Question 4. Lines 195 for temperature analysis mention "concentric domains with diameter of 40 mm and 100 mm, respectively" followed by "cell having 100nm diameter". Check it.
Answer 4 Here, we reorganized the phrases. The phrase from the page 6: ``In this temperature analysis, the abnormal and normal regions are two concentric domains with diameter of 40 mm and 100 mm, respectively (Figure 1)`` was moved before the paragraph which starts with ``4. In the following simulations, the space-time temperature values generated by the same magnetite MNPs doses having the cubic and spherical shapes in an magnetic field (f = 200kHz , H = 5 kA/m ) were computed from the equations (1), (6) and (8) for a malignant tissue having 40mm of diameter. At the center of the tumor-healthy tissue configuation a maximum clinical accepted concentration was Cmax = 10 mg/cm3``. In this part of paper will be described the temperature analysis for the tumoral – healthy tissues.
Question 5. With such an abundance of equations in the text, they should be drawn up more accurately. In particular, try to stick to a unified style. For example, in line 129 the same values are typed in italics and without it. In line 166 Ms must be subscripted. Correct it throughout the text.
Answer 5. We solved this point.
Question 6. Unify the character used to denote a range. Now in the text and tables there is a mixture of characters ÷ (in my opinion correct), - and —.
Answer 6 We solved this point.
Question 7. Figures 3 and 6 need to be improved and clarified.
7.1 There are no axes ticks, this makes the presentation of numerical information careless.
7.2 What is the meaning of the symbols in Figure 3? If these are the values where data was taken, why are they connected by a smooth line with a break feature around 10 nm where no data was taken?
7.3 For a cubic particle, the thermal front will not be completely spherical. What then is the meaning of r in Fig. 3 and r0 in Fig. 6? Is it correct to call this value the radius, because for a non-spherical front it will be different in different directions?
Answer 7. The temperature within tissues T(˚C) depends on the radial variable r (nm) and time t(s). The system of coordinates is located at the center of cell or tumoral – healthy tissues configuration. Due to this the radial variable can be positive or negative according to the location of coordinate system. We improved this figure.
In the fig 6 the r0 is the radius of the tissue which contains the therapeutic temperature values.
Question 8. Results obtained using a finite element method (FEM) in Comsol Multiphysics software. To assess the reliability of the obtained data, information about the finite element mesh is critical. This information needs to be added.
Answer 8. This paragraph was inserted in the paper: ``Discretization of the concentric tumoral-healthy tissues geometries was performed by finite element method (FEM) in Comsol Multiphysics software. The solutions of the equations (1), (6) and (8) were obtained using the extremely fine mesh which contains 50000 domain elements. The Time-Dependent Solver with implicit backward differentiation (BDF) method was used. This is an implicit solver which uses backward differentiation formulas with variable discretization order and automatic step-size selection. For quality of the solutions, the higher-order schemes are used. The variation of the time step in the course of the simulation can be followed in Convergence Plot``.
Thank you for your consistent recommendations and re-consideration.

Round 2
Reviewer 2 Report
The revised manuscript has improved and recommends for publication.
Author Response
Thank you for all recommendations.